# Experimental Infection of Embryonic Cells and Embryonated Eggs of Cockatiels (*Nymphicus hollandicus*) with Two Parrot Bornavirus Isolates (PaBV-4 and PaBV-2)

**DOI:** 10.3390/v14091984

**Published:** 2022-09-07

**Authors:** Elisa Wuest, Sarah Malberg, Jana Petzold, Dirk Enderlein, Ursula Heffels-Redmann, Sibylle Herzog, Christiane Herden, Michael Lierz

**Affiliations:** 1Clinic for Birds, Reptiles, Amphibians and Fish, Justus-Liebig-University Giessen, 35392 Giessen, Germany; 2Institute of Veterinary Pathology, Justus-Liebig-University Giessen, 35392 Giessen, Germany; 3Institute of Virology, Justus-Liebig-University Giessen, 35392 Giessen, Germany

**Keywords:** avian, psittacines, vertical transmission, species conservation, proventricular dilatation disease, central nervous system, nerval disease

## Abstract

Parrot bornavirus (PaBV) might be transmitted vertically. Cockatiel embryonic brain cells and embryonated eggs of cockatiels (ECE) were infected with PaBV-2 and PaBV-4. In embryonic brain cells, PaBV-2 and PaBV-4 showed no differences in viral spread despite the slower growth of PaBV-2 compared with PaBV-4 in CEC-32 cells. ECE were inoculated with PaBV-4 and 13–14 dpi, organs were sampled for RT-PCR, immunohistochemistry/histology, and virus isolation. In 28.1% of the embryos PaBV-4-RNA and in 81.3% PaBV-4-antigen was detected in the brain. Virus isolation failed. Division of organ samples and uneven tissue distribution of the virus limited the results. Therefore, 25 ECE were inoculated with PaBV-4 (group 1) and 15 ECE with PaBV-2 (group 3) in the yolk sac, and 25 ECE were inoculated with PaBV-4 (group 2) and 15 eggs with PaBV-2 (group 4) in the chorioallantoic membrane to use the complete organs from each embryo for each examination method. PaBV-RNA was detected in the brain of 80% of the embryos in groups 1, 2, 3 and in 100% of the embryos in group 4. In 90% of the infected embryos of group 1, and 100% of group 2, 3 and 4, PaBV antigen was detected in the brain. PaBV antigen–positive brain cells were negative for anti-neuronal nuclear protein, anti-glial fibrillary acidic protein, and anti S-100 staining. Virus was not re-isolated. These results demonstrated a specific distribution pattern and spread of PaBV-4 and PaBV-2 in the brain when inoculated in ECE. These findings support a potential for vertical transmission.

## 1. Introduction

*Psittaciform 1 bornavirus* (PaBV) has been described as cause of a fatal disease with gastrointestinal and neurological symptoms in various psittacine species (it fulfills Koch’s postulates) [1,2,3,4,5]. To stop the infection from spreading, it is crucial to know all possible transmission routes, but these remain unclear [6,7]. For breeding programs and species conservation, it is particularly important to have reliable data on potential vertical transmission. The demonstration of the parrot bornavirus (PaBV) antigen in the follicular cells of diseased birds points to the possibility of vertical transmission [8]. In another study involving the examination of 30 dead-in-shell embryos, PaBV RNA was found in the brains of two embryos [9]. The parents of one of the embryos did not exhibit PaBV serum antibodies, but it did shed PaBV RNA from the cloaca. For the second embryo, only the female parent shed PaBV RNA from the cloaca with no detectable PaBV serum antibodies, whereas the male parent did not shed PaBV RNA but showed PaBV serum antibodies. Both embryos had no histological alterations and virus isolation was not possible in either case, perhaps due to the limited sample quality due to autolysis [9]. Moreover, PaBV RNA was already detected in the embryos of four pairs of natural PaBV-infected sun conures (*Aratinga solstitialis*). In one 2-week-old chick of these pairs, PaBV RNA was also detected in the blood [10]. Moreover, two of the chicks of these pairs showed anti-PaBV antibodies at the beginning of the examination, but they were no longer detectable two weeks later. As some chicks of those infected parents remained negative for shedding PaBV RNA or seroconversion, the authors concluded that vertical transmission might be possible and suspected that after artificial incubation of eggs from infected parents, PaBV-negative chicks can be raised [10]. This was supported by investigating an amazon breeding flock where hand-raised chicks from PaBV-positive parents tested negative for PaBV RNA from crop and cloacal swabs [11]. This also raised doubts about the occurrence of vertical transmission. PaBV RNA has also been found in the eggs and brains of embryos from PaBV-positive parents of different psittacine species without the demonstration of infectious virus [12]. For canaries (*Serinus canaria forma domestica*), canary bornavirus (CnBV) RNA has been found in embryos of CnBV-positive parents. Notably, only the female birds with a persistent CnBV infection and high virus shedding laid eggs with CnBV-positive embryos. Females with intermittent CnBV RNA shedding and females of the contact group laid only CnBV RNA-negative eggs. Moreover, infectious virus was not detected in the embryos of these females [13]. In summary, according to the current available literature, virus isolation is not possible from avian bornavirus RNA-positive embryos regardless of the avian species and infections status of the parents. Therefore, the possibility of vertical transmission and natural infection of the embryo with PaBV remains unclear. 

In the present study, the potential of vertical transmission of PaBV-4 and PaBV-2 via different infection routes was investigated in cockatiels *(Nymphicus hollandicus*). For this purpose, the susceptibility for infection of embryonic cockatiel brain cells (CEB) and embryonic cockatiel fibroblasts (CEF) established from 10-day-old cockatiel embryos were investigated. These cells were inoculated with PaBV-4 or PaBV-2 isolates to investigate virus replication. Following the *in vitro* study, fertilized eggs of PaBV-free cockatiels were inoculated with either PaBV-4 or PaBV-2 and the embryos were investigated by reverse transcription–polymerase chain reaction (RT-PCR), histology, immunohistochemistry, and virus isolation for the presence of infection. We hypothesized that if virus propagation in the cell cultures fails and/or embryos cannot be infected by inoculating the virus, vertical transmission is very unlikely.

## 2. Material and Methods

### 2.1. Virus Isolates

The PaBV-4 isolate (Ps34) for inoculation originated from the brain of a scarlet macaw (*Ara macao*) that died from proventricular dilatation disease (PDD) and was successfully used in a study for experimental infection of cockatiels *(N. hollandicus*) [4]. For the inoculation of the embryonated eggs, 5 × 10^6^ median infectious tissue dose (TCID_50_)/mL of the inoculate was diluted with phosphate-buffered saline (PBS) to 2 × 10^3^ TCID_50_/0.02 mL per egg.

The PaBV-2 isolate (Ps39) originated from a cockatiel *(N. hollandicus*) that was presented in late stage PDD and was euthanized. The isolate had been used successfully in a previous infection trial [5]. For inoculation, 5 × 10^5^ TCID_50_/mL of the inoculate was diluted with PBS to 2 × 10^3^ TCID_50_/0.02 mL per egg.

### 2.2. In Vitro Inoculation of PaBV-4 and PaBV-2

#### 2.2.1. Embryonal Cockatiel Brain Cells (CEB)

The brains of 10-day-old cockatiel embryos were removed with a 10-mL syringe that contained 2 mL of Dulbecco’s Modified Eagle’s medium (DMEM, Gibco, Invitrogen, Life Technologies Ltd., Paisley, UK) with 10% fetal calf serum (FCS) through a hypodermic needle (20 G × 1.5 inches). The brain suspension was added to 1 mL of DMEM with 10% FCS onto chamber slides (Nunc A/S, Roskilde, Denmark). The medium was changed after 2 days of incubation at 37 °C. Inoculation with PaBV-4 or PaBV-2 was performed after the monolayer showed 75% confluence. The cells were inoculated with 10-fold serial dilutions in medium containing 10% FCS. Every dilution was incubated on chamber slides for 6 days at 37 °C. 

#### 2.2.2. Embryonal Cockatiel Fibroblast Cells (CEF)

The head, extremities, and organs of 10-day-old cockatiel embryos were removed. The empty body cavity was cut into small pieces and incubated with enzymes (0.025% trypsin-EDTA) at 37 °C with slow shaking for 30 min. The cell suspension was filtered through gauze and centrifuged at 200× *g* for 10 min. The pellet was diluted with 10% FCS and placed onto chamber slides. The medium was changed after 2 days. Inoculation was performed as described above for CEB.

To identify viral replication on CEB and CEF, the cells were fixed with acetone and incubated with a polyclonal rat anti-Borna disease virus (BoDV) serum with cross-reactivity to PaBV [14]. The antigen–antibody reaction was detected with a goat anti-rat IgG linked to fluorescein isothiocyanate (FITC; Dianova GmbH, Hamburg, Germany) diluted 1:400 in immunoblock buffer. The cells were evaluated using a fluorescence microscope (Olympus BX60, Olympus Deutschland GmbH, Hamburg, Germany) at 492 nm and 400× total magnification. There is brilliant granular fluorescence in the nucleus of an infected cell.

### 2.3. In Vivo Inoculation of PaBV-4 and PaBV-2

In a first study (trial A), PaBV-4 was inoculated into the yolk sac of 32 fertilized cockatiel eggs between the third and the fifth day of incubation after shell disinfection using 3% hydrogen peroxide, which is an appropriate disinfectant for the egg shell against PaBV [11]. Two additional eggs injected with culture medium only served as mock controls. The eggs were incubated at 37.4 °C and 60% humidity until day 12–14 post-infection (p.i.), and each organ of each embryo was divided for RT-PCR, histology, immunohistochemistry, and virus isolation. The division of the organs of one embryo was not ideal (see Section 4). Therefore, a follow-up study (trial B) was conducted with four groups. 

In trial B, groups 1 and 2 consisting of 25 embryos each were inoculated with 2 × 10^3^ TCID_50_/0.02 mL of PaBV-4 per egg in the yolk sac (group 1) on day 5 or in the chorioallantoic membrane (group 2) on day 5–7 of incubation. Each group included two more eggs as mock controls as described above. The same study design was conducted with groups 3 and 4, where 15 embryos in each group were inoculated with 2 × 10^3^ TCID_50_/0.02 mL per egg of PaBV-2 into the yolk sac (group 3) or the chorioallantoic membrane (group 4) with two additional mock controls per group. Similarly to trial A, inoculated eggs were incubated for 12–14 days p.i. The brain and organs (proventriculus, gizzard, liver, and kidney) of 10 embryos each for groups 1 and 2, 5 embryos each for groups 3 and 4, and one mock control were exclusively examined through RT-PCR for the presence of viral RNA. Furthermore, 10 embryos each for groups 1 and 2 and 5 embryos each for groups 3 and 4 were taken for histological and immunohistochemical examinations. In addition, one mock control and five embryos from each group were used for virus isolation.

### 2.4. RT-PCR

RNA was isolated by using the RNeasy Mini Kit (QIAGEN, Hilden, Germany) according to the manufacturer’s instructions. Total RNA was reverse transcribed by using random hexamer primers. PaBV-4 was detected using the protocol described by Honkavuori et al. [2] with the primer “1034–1322”. PaBV-2 was detected as described by Piepenbring et al. [5] with the forward and reverse primers and a probe (forward primer: 5′-CAGACAGCACGTCGAATGAGG-3′; probe: 6-FAM-5′-AGGTCTCCAAGAAGGAAGCGA-3′-TMR; reverse primer: 5′-AGTTAGGGCCTCCCTGGGTAT-3′) as previously described. Samples were considered positive if the melting curve demonstrated the typical appearance of the positive control.

### 2.5. Histology

Histology was performed by using hematoxylin–eosin staining [15] at the Institute of Veterinary Pathology, Justus-Liebig-University Giessen. Microscopic evaluation was performed at 100×, 200×, and 400× magnification for the presence of any lesions. 

### 2.6. Immunohistochemistry

Immunohistochemistry was performed according to the avidin–biotin complex method, as described elsewhere [16]. The chromogen diaminobenzidine (DAB) (Fluka Chemikalien, Taufkirchen, Germany) was used to label PaBV antigen–antibody complexes in tissue sections; they produced a brown reaction product. A polyclonal antibody directed against p24, a phosphoprotein of BoDV-1, with proven cross-reactivity to PaBV, was used [4,14]. All positive cells in one field of view (200× total magnification) were counted in each of three brain sections: the cerebellum, the cerebrum with ventricles, and the central cerebrum without ventricles. 

To identify PaBV antigen-positive cells, the same ABC method or the PAP method was applied. Anti-neuronal nuclei (NeuN, a neuronal marker), rabbit anti-glial fibrillary acidic protein (GFAP; an astroglia marker), and rabbit anti-S-100 PAP (a glia and neuronal marker) were used to identify the cells. Tissue slides of six embryos (two from each of groups 1 and 2, one from each of groups 3 and 4) were used for this purpose.

### 2.7. Virus Isolation in the CEC-32 Cell Line

Infectivity assays were performed as described elsewhere [17]. The head, organs, and yolk of the embryos were homogenized, and a 10% organ suspension in DMEM with 2% FCS was compounded and decomposed with ultrasound and centrifuged at 1000× *g* for 10 min. Then, 100 µL of the supernatant was added to semiconfluent CEC-32 cells in 24-well tissue culture plates. After virus absorption, fresh DMEM with 10% FCS was added to the cells and they were incubated. Cells were passaged after 5 days. Indirect immunofluorescence assays were performed on multi-test slides after every cell passage. In total, all cell cultures were passaged three times.

### 2.8. Virus Isolation in Embryonic Cockatiel Cells

Processing was performed as described in Section 2.7 for CEC-32-cells. The tissue suspensions were incubated on CEB or CEF in chamber slides. Evaluation was performed after 5 days of incubation with the indirect immunofluorescence assay as described above. The cells were not passaged.

### 2.9. Statistical Analysis

The relative frequency and the 95% confidence interval was calculated for each group in both trials. “Comparison of occurrence” was only calculated for trial B, comparing the route of infection and the used isolate for the RT-PCR and immunohistochemistry results. A *p*-value < 0.5 was considered to indicate a significant difference between the inoculation routes as well as the used isolates. To determine potential significant differences between virus distribution in different brain areas, immunochemistry results from trial B for the cerebrum with and without ventricles as well as the cerebellum were compared using the Mann–Whitney U test. 

## 3. Results

### 3.1. In Vitro Inoculation of PaBV-4 and PaBV-2

Both PaBV isolates could be propagated on CEB but not on CEF. The growth results on CEB (PaBV-4: 5 × 10^6^ TCID_50_/mL; PaBV-2: 5 × 10^5^ TCID_50_/mL) were similar to the titers that were reported for PaBV-4 and PaBV-2 in earlier studies on CEC-32 cells [4,5]. However, in contrast to the earlier studies, PaBV-2 demonstrated significantly faster growth on CEB compared with CEC-32 cells. In addition, pleomorphic cells with multiple nuclei were seen in the CEB culture; they were mainly infected with PaBV-2 and PaBV-4 (Figure 1 and Figure 2). 

### 3.2. In Vivo Inoculation of PaBV-4 and PaBV-2

#### 3.2.1. Trial A: Yolk Sac Inoculation with PaBV-4

Overall, 9 of 32 embryos (28.1%) had PaBV-4 RNA and 26 of 32 embryos (81.3%) had PaBV antigen in the brain. PaBV antigen was detected in eight of nine embryos with PaBV-4 RNA in the brain. For the remaining embryo, there was not enough tissue left for immunohistochemical examination. Histological examination revealed no pathological alterations. Virus isolation was not possible in any of the examined embryos. The mock controls remained negative in all examinations (Table 1).

#### 3.2.2. Trial B: Yolk Sac and Chorioallantoic Membrane Inoculation with PaBV-4 or PaBV-2

PaBV RNA was detected in the brains of 8 of 10 embryos (80%) of group 1, 8 of 10 embryos (80%) of group 2, 4 of 5 embryos (80%) of group 3, and all embryos of group 4. All other organs as well as the controls remained negative (Table 2).

PaBV antigen was detected in the brain of 9 of 10 embryos (90%) of group 1, and all embryos of groups 2, 3, and 4. However, PaBV antigen was not detected in the other organs of any group. PaBV antigen was most prevalent in the cerebellum *(arbor vitae*) and in brain areas with ventricles compared with the central region of the cerebrum without ventricles (Figure 3). 

For group 1, there were significantly more PaBV antigen-positive cells in the cerebellum than the cerebrum without ventricles (U = 17.5, *p* = 0.01552). There was not a significant difference in the PaBV antigen distribution between the cerebellum and the cerebrum with ventricles (U = 47.5, *p* = 0.88076). There was a significantly higher number of PaBV antigen-positive cells in the cerebrum with ventricles than the cerebrum without ventricles (U = 16, *p* = 0.0114) (Figure 4). 

In group 2, there were significantly more PaBV-4 antigen-positive cells in the cerebellum than the cerebrum without ventricles (U = 2, *p* = 0.00034). There was not a significant difference in the PaBV antigen distribution between the cerebellum and the cerebrum with ventricles (U = 44.5, *p* = 0.70394). There were significantly more PaBV antigen-positive cells in the cerebrum with ventricles than the cerebrum without ventricles (U = 8.5, *p* = 0.00194) (Figure 5). 

In group 3, there were significantly more PaBV antigen-positive cells in the cerebellum than the cerebrum without ventricles (U = 1, *p* = 0.02144). There was not a significant difference in the PaBV antigen distribution between the cerebellum and the cerebrum with ventricles (U = 7, *p* = 0.29834). Moreover, there was not a significant difference between the cerebrum with and without ventricles (U = 5.5, *p* = 0.17384) (Figure 6). 

In group 4, there was no difference in the number of PaBV antigen-positive cells between the cerebellum and the cerebrum without ventricles (U = 5, *p* = 0.1443) or between the cerebellum and the cerebrum with ventricles (U = 12, *p* = 1). However, there were significantly more PaBV antigen-positive cells in the cerebrum without ventricles than the cerebrum with ventricles (U = 0, *p* = 0.01208) (Figure 7).

None of the infected cells in the brain were positive for NeuN, GFAP, and/or S-100. Regardless of the group, none of the embryos showed histological alterations in any organ. It was not possible to isolate PaBV from any inoculated embryos or from the control group. 

There were no significant differences for the different PaBV isolates (*p* = 1 (RT-PCR) and *p* = 1 (immunohistochemistry)) and the different inoculation routes (*p* = 0.6053 (RT-PCR) and *p* = 1 (immunohistochemistry)) related to the susceptibility of the embryos for any of the infection routes applied.

## 4. Discussion

In 2008, *Psittaciform 1 bornavirus* was described as the cause of a fatal disease with gastrointestinal and neurological symptoms [1,2,3,4,5]. Experimental trials excluded oral/nasal infection as the route of transmission and suggested wounds as the portal of entry [6,7]. Additional natural transmission routes remain unclear. However, this information is required to stop the infection from spreading [6]. Especially for breeding and species conservation programs, it is crucial to know whether vertical transmission may occur. Thus, our present study focused on the potential of PaBV vertical transmission. Prior to the *in vivo* trials, different embryonic cockatiel cells were inoculated with PaBV-4 and PaBV-2 to assess their susceptibility. Viral replication on CEB provided the first hints that cockatiel embryos can indeed be infected with PaBV-4 and PaBV-2. Both PaBV isolates showed similar growth rates on CEB, which is contrary to published data that PaBV-2 grows slower than PaBV-4 in CEC-32 cells [18,19]. This may be due to the fact that the PaBV-2 isolate originated from a cockatiel and was therefore already more adapted to these cells. Piepenbring et al. [5] also reported such a potential adaption. Infection with PaBV-2 led to stronger clinical occurrence than infection with PaBV-4. Furthermore, pleomorphic cells were detected in the CEB culture and preferably those cells were infected. It was not possible to identify the origin of these cells, so further investigation is recommended. Both PaBV isolates also showed significant neurotropism in the embryonic cell culture: Viral replication were detected on CEB but not on CEF. This neurotropism is well described in adult rodents [20]. Only in immunoincompetent mammals can BoDV-1 also spread to other organs [17]. However, it remains unclear why both isolates grow on CEC-32 cells because this cell line originated from fibroblasts. One reason might be that CEF are primary cells, while CEC-32 cells have been immortalized and propagated for many years, with changes in cellular expression patterns. However, this requires further investigation. Interestingly, the cell culture results were supported by the *in vivo* trials. After inoculating the yolk sac or chorioallantoic membrane with PaBV, PaBV RNA and antigen were only detected in the brain regardless of the virus isolate used. There were no significant differences between the inoculation routes or the different inoculates. 

Trial A was performed by inoculating the yolk sac of 32 embryonated cockatiel eggs with PaBV-4. The brain of each embryo was divided and subjected to all examination methods (PCR, histology, immunohistology, and virus isolation), which resulted in a small sample size for each examination. Given the inconsistent results (28.1% PaBV RNA-positive embryos vs. 81.3% PaBV antigen-positive embryos), this methodology was considered to be not ideal. Trial B demonstrated the uneven distribution of PaBV in the brain, which could have led to the variable results in trial A. The discrepant RT-PCR and antigen results in trial A did not allow us to determine whether virus isolation failed due to the small sample size or a low titer of live virus or even a non-live virus. Hence, trial B was performed with a larger number of inoculated embryos and using each organ for one examination method. The differences in the results of trial A and B clearly demonstrate that the division of samples may lead to inconsistent or even false results, especially if the virus is unevenly distributed in the tissue. Therefore, the approach used for trial A is not recommended in future studies examining small embryos.

In trial B, PaBV antigen was found especially in the *arbor vitae* of the cerebellum (*corpus medullare cerebelli*) and in the cerebrum close to the ependymal layered ventricle and only rarely at the center. Ependymal cells are in contact with neurons and to liquor (cerebral fluid), which may represent two potential routes of virus expansion within the brain. This distribution pattern was not seen in trial A, most likely due to the subdivision of the brain of each embryo for the different examination methods. The demonstration of viral antigen in cells located at the base of a ventricle is an especially interesting finding. These cells are located close to the ependymal layer and could be stem cells, as has been described for humans [21]. Different antibodies were used to identify these cells. Although these antibodies have been established for adult cockatiel brain cells, they were unable to identify the origin of the embryonic brain cells. This might be due to a lack of cross-reactivity because these antibodies are generated against mammalian antigens or a lack of expression of these markers in immature embryonic cells. Whether those cells represent the same cell type as the pleomorphic cells found in the CEB culture warrants further investigation. 

Neither PaBV antigen nor RNA was found in the other organs of the embryos regardless of the infection group. This finding indicates tropism to cells of the nervous tissue, consistent with the *in vitro* experiments. It is unclear whether centripetal viral spread can occur after successful infection of the brain in embryos or even after hatching, as has been described for the infection of adult cockatiels. In particular, in adult cockatiels, other tissues are also affected, especially if clinical disease occurs [4,5]. Kerski et al. [10] described that at the beginning of embryogenesis, PaBV RNA was only detected in the brain, but later it was also found in the organs of the embryo and in the blood of a hatched chick. Whether infected embryos spread and shed PaBV via other organs at a later time or might even be inapparently infected or develop clinical signs requires additional investigation. In particular, the possibility of lifelong PaBV carriers after vertical transmission—if proven—needs further attention, especially because the natural transmission routes of PaBV are only partly known [6,7], and carriers can be a potential danger to rare species. Last but not least, it should be considered that the low amount of virus detected might be an abortive infection so that the embryo is able to control or even eliminate the virus at a later point. This mechanism could reduce or even prevent vertical transmission.

As described previously [9,10,12,13], the virus could not be isolated from any of the infection groups, regardless of the virus isolate or infection route. This could be due to a low virus load and the restriction of virus infection to certain brain areas 14 days p.i. The Ct values in the brain showed a relatively low RNA load with an arithmetic average of 28.8. In addition, the slow spread of PaBV to the brain could represent another reason for the low virus load; the virus might have just arrived, a phenomenon that resembles mammalian bornaviruses [22]. Whether the viruses can undergo successful replication in the cerebellum during subsequent embryonic development also needs to be analyzed by examining artificially inoculated and naturally infected embryos. It might be possible that the PCR detected residual virus from inoculation, explaining the high Ct values. However, this seems unlikely because the virus was inoculated in the yolk sac or the chorioallantoic membrane but only the brain was positive for PaBV RNA. If the detected viral RNA had originated from the inoculated virus, other organs would also have been positive, especially those closer to the yolk sac. In addition, PaBV antigen was found in brain cells, indicating PaBV infection. 

In conclusion, PaBV RNA and antigen were successfully detected after PaBV inoculation in the embryonated egg. The distribution of PaBV antigen in the brain, especially in the *arbor vitae* of the cerebellum and to a lesser extent in the cerebrum, suggest a preference for certain brain areas in the developing embryo. Whether this might result in a persistent infection with or without further spread within the brain or other peripheral organs, viral shedding, and the occurrence of infectious virus, or whether this is an abortive infection, needs to be investigated further. This will contribute to being able to reliably assess the potential of vertical transmission. In particular, the possibility of lifelong PaBV carriers after vertical transmission should be addressed as a potential hazard for spread of the infection. The present study supports the possibility of the vertical transmission of PaBV in psittacines. However, vertical transmission still needs to be proven by isolating infectious virus from an embryo after natural transmission as well as infected virus-positive hatched chicks.

## Figures and Tables

**Figure 1 viruses-14-01984-f001:**
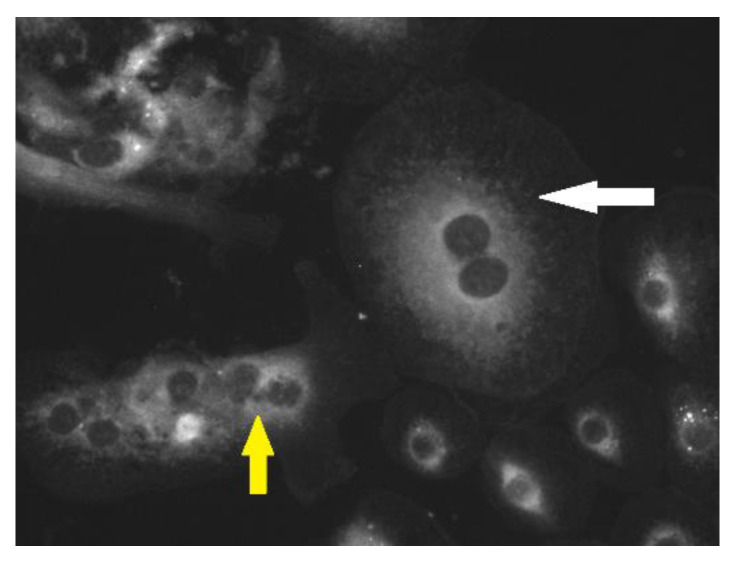
Uninfected embryonic cockatiel brain cells, including a large pleomorphic cell with two nuclei (white arrow) alongside smaller cells with multiple nuclei (yellow arrow).

**Figure 2 viruses-14-01984-f002:**
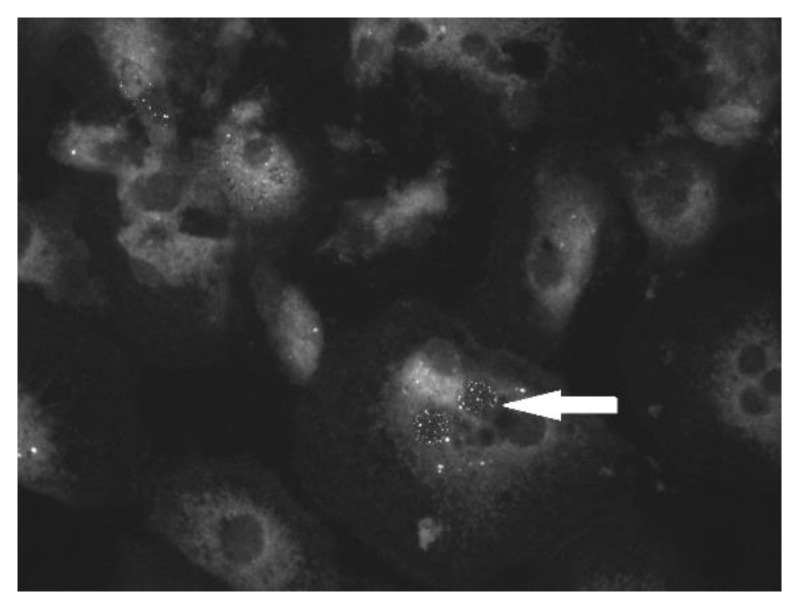
Indirect immunofluorescence assay: PaBV-4-infected embryonic cockatiel brain cells with granules in the nuclei (white arrow).

**Figure 3 viruses-14-01984-f003:**
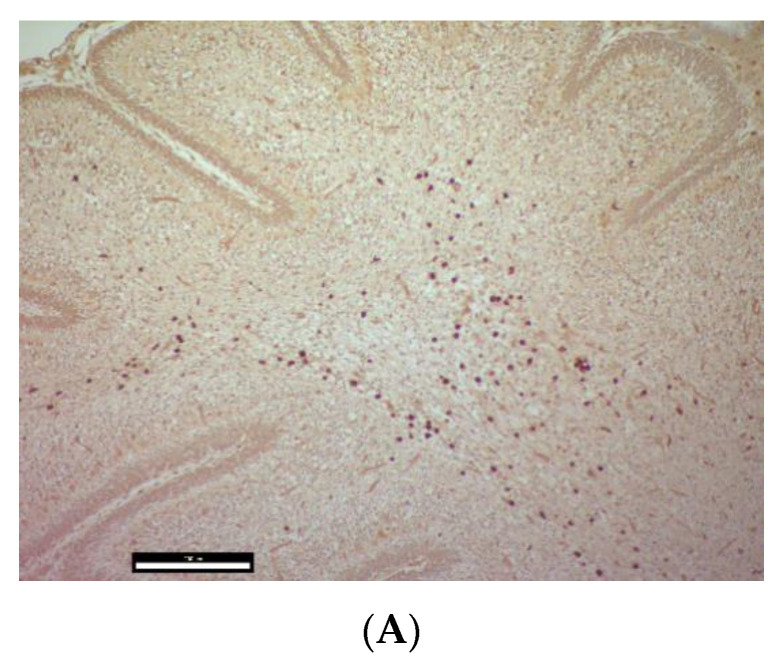
Parrot bornavirus 4 (PaBV-4) antigen detection in the brain of trial B embryos. (**A**) Cerebellum with PaBV antigen-positive brain cells, DAB staining in the *arbor vitae* (embryo DAB_PABV-2_01, 100× total magnification, size bar 200 µm). (**B**) Cerebrum with ventricle, positive brain cells by DAB staining at the base of ependymal cells (Embryo DAB_PABV-4_DO01, 200× total magnification, size bar 100 µm). (**C**) Cerebrum without ventricle with only a few positive cells visualized by DAB staining (embryo DAB_PABV-2_01, 100× total magnification, size bar 200 µm).

**Figure 4 viruses-14-01984-f004:**
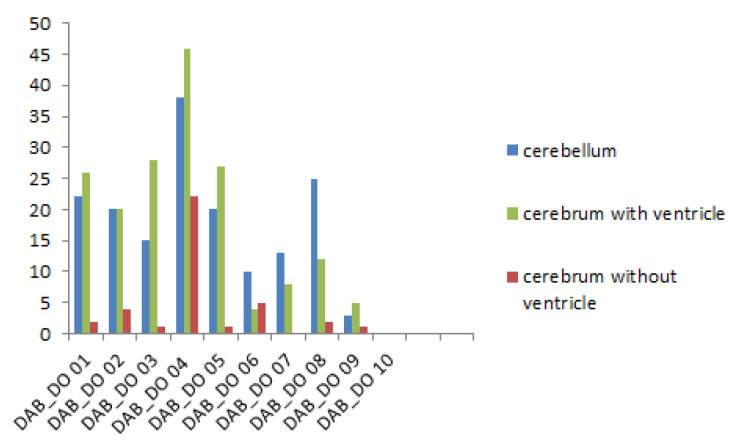
Parrot bornavirus (PaBV) antigen distribution in the brain of trial B group 1 (PaBV-4 inoculation in the yolk sac). The *y*-axis shows the number of PaBV antigen-positive cells counted in one slide (200× total magnification). PaBV antigen was mainly found in the cerebellum and cerebrum with ventricles, and much less frequently in the cerebrum without ventricles. The *x*-axis shows each individual embryo (*n* = 10, laboratory number).

**Figure 5 viruses-14-01984-f005:**
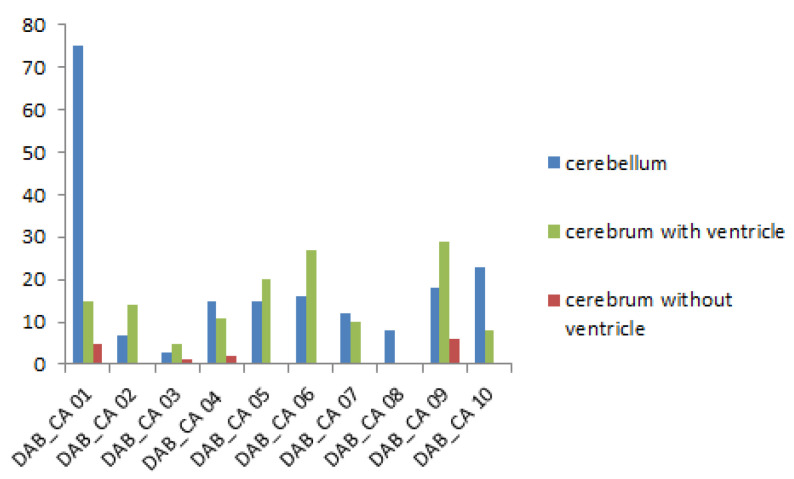
Parrot bornavirus (PaBV) antigen distribution in the brain of trial B group 2 (PaBV-4 inoculation in the chorioallantoic membrane). The *y*-axis shows the number of PaBV antigen-positive cells counted in one slide (200× total magnification). The PaBV antigen distribution similar to trial B group 1. The *x*-axis shows each individual embryo (*n* = 10, laboratory number).

**Figure 6 viruses-14-01984-f006:**
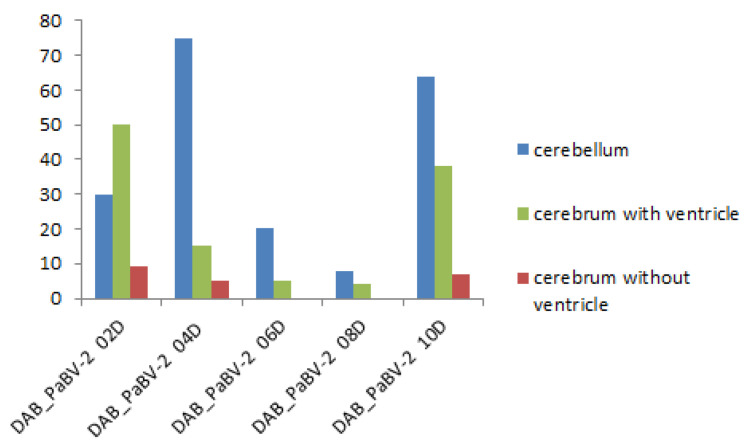
Parrot bornavirus (PaBV) antigen distribution in the brain trial B group 3 (PaBV-2 inoculation in the yolk sac). The *y*-axis shows the number of PaBV antigen-positive cells counted in one slide (200× total magnification). The PaBV antigen distribution is similar to trial B groups 1 and 2. There was no significant difference between the cerebrum with and without ventricles. The *x*-axis shows each individual embryo (*n* = 5, laboratory number).

**Figure 7 viruses-14-01984-f007:**
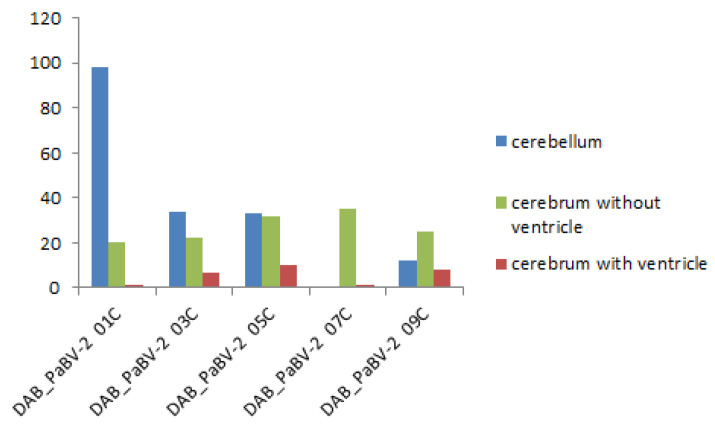
Parrot bornavirus (PaBV) antigen distribution in the brain of trial B group 4 (PaBV-2 inoculation in the chorioallantoic membrane). The *y*-axis shows the number of PaBV antigen-positive cells counted in one slide (200× total magnification). The PaBV antigen distribution is similar to trial B group 3. There was no significant difference between the cerebellum and the cerebrum without ventricles and between the cerebellum and the cerebrum with ventricles. There was a significant difference between the cerebrum with and without ventricles. The *x*-axis shows each individual embryo (*n* = 5, laboratory number).

**Table 1 viruses-14-01984-t001:** Overview of the trial A results (yolk sac inoculation with parrot bornavirus 4 (PaBV-4)).

No.	PaBV RNA Ct Value in the Brain	PaBV RNA Ct Value in the Proventriculus, Gizzard, Liver, and Kidney	PaBV Antigen in the Cerebellum	PaBV Antigen in the Cerebrum	PaBV Antigen in the Proventriculus, Gizzard, Liver, and Kidney
	23.34	-	+	-	-
Egg 02	-	-	+	-	-
Egg 03	-	-	No material	No material	-
Egg 04 contr.	-	-	-	-	-
Egg 05	31.03	-	No material	No material	-
Egg 06	-	-	+	No material	-
Egg 07	-	-	No material	+	-
Egg 08	-	-	+	+	-
Egg 09	-	-	No material	No material	-
Egg 10	-	-	+	+	-
Egg 11	30.59	-	+	-	-
Egg 12 contr.	-	-	-	-	-
Egg 13	-	-	-	+	-
Egg 14	24.75	-	+	+	-
Egg 15	-	-	No material	+	-
Egg 16	-	-	No material	+	-
Egg 17	-	-	+	+	-
Egg 18	-	-	No material	+	-
Egg 19	-	-	+	-	-
Egg 20	-	-	+	-	-
Egg 21	-	-	+	- Less material	-
Egg 22	-	-	+	No material	-
Egg 23	31.03	-	+	+	-
Egg 24	-	-	+	+	-
Egg 25	-	-	+	-	-
Egg 26	-	-	No material	-	-
Egg 27	-	-	+	No material	-
Egg 28	-	-	+	No material	-
Egg 29	37.36	-	No material	-	-
Egg 30	-	-	-	-	-
Egg 31	30.84	-	+	-	-
Egg 32	-	-	+	+	-
Egg 33	34.88	-	+	-	-
Egg 34	26.48	-	+	No material	-

For PaBV RNA detection, a Ct value means the sample was positive; “-” means the sample was negative. For PaBV antigen detection, “+” means positive and “-” means negative. Contr. = control.

**Table 2 viruses-14-01984-t002:** Overview of PaBV RNA results for trial B yolk sac or chorioallantoic membrane inoculation with parrot bornavirus 4 (PaBV-4) or PaBV-2.

Group 1 Yolk Sac Inoculation with PaBV-4 (Laboratory Number of Embryo)	Ct Value in the Brain	Ct Value in the Proventriculus, Gizzard, Liver, and Kidney
PCR_DO 01	21.17	-
PCR_DO 02	30.47	-
PCR_DO 03	29.56	-
PCR_DO 04	-	-
PCR_DO 05	21.09	-
PCR_DO 06	28.75	-
PCR_DO 07	38.90	-
PCR_DO 08	21.78	-
PCR_DO 09	21.81	-
PCR_DO 10	-	-
PCR_DO 11 control	-	-
**Group 2 chorioallantoic membrane with PaBV-4**		
PCR_CA 01	23.02	-
PCR_CA 02	21.87	-
PCR_CA 03	24.33	-
PCR_CA 04	22.42	-
PCR_CA 05	31.46	-
PCR_CA 06	31.10	-
PCR_CA 07	32.48	-
PCR_CA 08	-	-
PCR_CA 09	25.07	-
PCR_CA 10	-	-
PCR_CA 11 control	-	-
**Group 3 yolk sac inoculation with PaBV-2**		
PCR_PaBV-2 02 DO	26.43	-
PCR_PaBV-2 04 DO	25.23	-
PCR_PaBV-2 06 DO	39.63	-
PCR_PaBV-2 08 DO	26.41	-
PCR_PaBV-2 10 DO	-	-
PCR_PaBV-2 12 DO control	-	-
**Group 4 chorioallantoic membrane inoculation with PaBV-2**		
PCR_PaBV-2 01 CA	31.16	-
PCR_PaBV-2 03 CA	38.11	-
PCR_PaBV-2 05 CA	36.94	-
PCR_PaBV-2 07 CA	29.66	-
PCR_PaBV-2 09 CA	29.90	-
PCR_PaBV-2 11 CA control	-	-

A Ct value means the sample was positive for PaBV RNA; “-” means the sample was negative.

## Data Availability

Not applicable.

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
