# Peer review of "Experimental Infection of Embryonic Cells and Embryonated Eggs of Cockatiels (Nymphicus hollandicus) with Two Parrot Bornavirus Isolates (PaBV-4 and PaBV-2)"

_viruses, 2022, doi:10.3390/v14091984_

Round 1

Reviewer 1 Report

In the manuscript by Wuest et al., the authors present data of experimental infection with two parrot Bornavirus isolates (PaBV-4 and PaBV-2), notably by assessing the permissivity of different cell lines of embryonic origin. The underlying rationale was to assess a possible vertical transmission of these viruses , although I am not entirely convinced that the proposed methodology would really address this issue (see below).

Considering our lack of knowledge of the biology of these new bornaviruses from parrots and their possible devastating consequences for psittacines due to proventricular dilatation disease, it is clear that more fundamental knowledge is warranted. In that respect, this study is justified, notably because it proposes to assess infection using primary cells or direct infection of eggs, which are relevant systems. Unfortunately, however, this work suffers from several limitations and needs to be extensively revised to reach a better standard.

1)    I have difficulties in presenting this work as a "first step" to examine the possibility of vertical transmission. Assessing infection of embryonic cells, or event directly infecting eggs, does not bring any evidence pro or against vertical transmission. It only informs about susceptibility to infection. Testing vertical transmission would require experimental infections of females and next assessing whether eggs could be positive. These aspects were addressed in the Introduction (lines 50 to 70), but the reasoning and presentation of this part is quite obscure and difficult to follow (notably lines 65 to 70, where I could not even understand the sentences and the point made by the authors). The authors should clarify their introduction and better explicit the rationale of their work, which mainly aims at assessing the susceptibility of embryonic tissues to infection.

2)    I was also disappointed by the presentation of the results, which appears as a collection of experiments (nearly an excerpt of a lab book) with no clear efforts to perform a critical and synthetic analysis of the data. For instance, if the methodology used for trial A turned out to be not adapted for the study, what is the rationale for showing it? How can we critically analyze these results with respect to results of trial B?

3)    I have not understood the statistical methodology used, first to analyze the immunohistochemistry results and also to combine IHC and RT-PCR results. This part should be better explained and detailed, notably by adding number of experimental and technical replicates for each experiment, etc.

4)    Importantly, the style, English and presentation of the manuscript should be carefully revised and corrected. There is a near total lack of punctuation (and the rare commas in the text are often misplaced), which makes reading very difficult. There are also numerous typos or unknown/misspelled words (serumantibodies, fluorochrom, Isothiocyanat, gliacell, FKS, Wenig… to only cite a few). This doesn't help to appreciate the actual quality of the work.

5)    Regarding RT-PCR Ct values upon experimental infection, I have difficulties in considering that Ct values over 30 are indeed a proof of infection and not residual inoculum. Could the authors define their cut-off value and interpretation?

6)    The authors should be aware that the protocol used for producing embryonic (and not embryonal) cockatiel brain cells (called CEB in the text), will only yield astrocytes and no neurons. Indeed, brains cells should rather be cultivated in a medium prone to favor neuronal survival (such as Neurobasal) and without Fetal Calf Serum (FCS and not FKS as written), which kills neurons, using instead supplements such as B27 or N2. Moreover, preparing the tissue by mechanical disruption (through a needle), rather than by mild enzymatic dissociation (using Papain) will cause extensive axotomy and neuronal death, leaving only the astrocytes, which are far more resistant. As a matter of fact, cells shown in Fig 1 clearly look like glial cells, and none has a neuronal morphology. Did the authors try co-labeling with GFAP in these cells?

Reviewer 2 Report

In this manuscript, the authors examined whether parrot bornavirus 2 (PaBV-2) and parrot bornavirus 4 (PaBV-4) spread via vertical transmission by inoculating these viruses into embryonal cockatiel cells and fertilized cockatiel eggs. Both viruses successfully established persistent infection in embryonal cockatiel brain cells. In addition, although infectious virus was not isolated, viral RNAs and antigens were detected especially in the cerebellum and cerebrum with ventricles of embryos that were inoculated with either PaBV-2 or PaBV-4. Overall, while further investigation is still necessary, this manuscript demonstrates the possibility of vertical transmission of parrot borna viruses. This work is a welcome addition to this field, but some minor corrections and revisions listed below are desirable before publication.

1.      Some words are written in German language, such as “und” in line 82 and “wenig” in table 1. Please read the manuscript carefully again to correct these word to English.

2.      Fig. 1 legend; describe virus genotype used for this experiment. PaBV-2?

3.      Fig. 1 and 2; indicate which cells are mentioned in the figure legends with arrows or other means.

4.      In in vitro experiments, both PaBV-2 and PaBV-4 isolates can infect to CEC-32 cells but not to CEF cells, despite both cells derived from embryonal fibroblast. Why?

5.      Have polymorphic cells with multiple nuclei been seen in other cells infected with PaBVs, such as CEC-32 cells?

6.      In trial B, how long were the eggs incubated after virus inoculation? Does the period affect the levels of viral RNA and antigen? Please discuss.

7.      Fig. 3, cannot see bar size.

8.      Figs. 4 to 7, The graphs should be processed to be more understandable, not the graphs themselves generated from Excel. For example, describe what the numbers on the vertical axis indicate. Also, indicate in the graph which groups were significantly different in the graphs compared to which. The sample names for the X axis are also confusing because they are different from the names that appear in the text.

9.      In in vivo experiments, detection rate of PaBV-4 RNA differed greatly between trial A and trial B. Could you explain the reason for this difference?

Round 2

Reviewer 1 Report

This revised version of the manuscript by Wuest et al. has been significantly improved.

The authors have to be commended on their reactivity and quick investment to correct the initial version and address all issues raised.

The use of a professional academic service for revising the text has indeed been a very  important improvement, because the rationale and analysis of the data is now well elaborated and convincing. The rationale of the study as a first step to address the possibility of vertical transmission is much clearer and critically presented. Likewise, the rationale for presenting both trials is clear and I now adhere to the choice made by the authors, as it may indeed be useful for future studies, in particular when working with small numbers of samples.

The statistical methods are also clearer and better presented.

One minor comment is that it would have been very useful to also have access to a "clean copy" of the revised manuscript, because it is sometimes very complicated to really appreciate the text under its final form with all the corrections still present under the "track changes" mode. Therefore, I may have missed some minor remaining typos.

I have also minor comments regarding the figures: in the legends for Fig 1 and Fig 2, the authors refer to "arrows" present in the pictures, but I could not see any; in the legends of Fig 3, rather than indicating "original magnification 100X", which is not very informative, the authors could indicate the size of the scale bar (there is a number above it, but it does not read well in the figure); finally, the "Excel" formatting of the graphs is not very nice and could be presented in a more "standard" manner.

Author Response

Reviewer 1 had the following comments:

This revised version of the manuscript by Wuest et al. has been significantly improved. The authors have to be commended on their reactivity and quick investment to correct the initial version and address all issues raised. The use of a professional academic service for revising the text has indeed been a very  important improvement, because the rationale and analysis of the data is now well elaborated and convincing. The rationale of the study as a first step to address the possibility of vertical transmission is much clearer and critically presented. Likewise, the rationale for presenting both trials is clear and I now adhere to the choice made by the authors, as it may indeed be useful for future studies, in particular when working with small numbers of samples.

Thank you very much. We are delighted that the reviewer is now satisfied with our manuscript

The statistical methods are also clearer and better presented.

Thanks- no action needed

One minor comment is that it would have been very useful to also have access to a "clean copy" of the revised manuscript, because it is sometimes very complicated to really appreciate the text under its final form with all the corrections still present under the "track changes" mode. Therefore, I may have missed some minor remaining typos.

We agree- but that was the request of the journal- so no action needed from us

I have also minor comments regarding the figures: in the legends for Fig 1 and Fig 2, the authors refer to "arrows" present in the pictures, but I could not see any; in the legends of Fig 3, rather than indicating "original magnification 100X", which is not very informative, the authors could indicate the size of the scale bar (there is a number above it, but it does not read well in the figure);

We do not know why the manucript still contain the old figures without the arrows. The new figures with arrows were submitted in the last revision. Editor: Please see the comments from us in the manuscript which figures need to be deleted and which should be exchanged. All have already been submnitted with the revision- also- there is already a scale bar in the figure- so no action needed from our side

finally, the "Excel" formatting of the graphs is not very nice and could be presented in a more "standard" manner.

The presentation is done in a standard manner and we are unable to submitt it in a differernt format. As this was accepted all the time before and also by the other reviewer- we think that this is ok as it is- so no action needed from our side.

Therefore the manuscript which we have alreade submitted as a first revision is still the latest version. We hope that publication can now be processed.

Reviewer 2 Report

The revised version of manuscript has made all necessary revisions and improved the quality of this manuscript.

Author Response

Comments Reviewer 2:

The revised version of manuscript has made all necessary revisions and improved the quality of this manuscript.

Thank you very much. No action needed from our side.